# Traffic Regulator Detection and Identification from Crowdsourced Data—A Systematic Literature Review

**Stefania Zourlidou \* and Monika Sester**

Institute of Cartography and Geoinformatics, Leibniz University, Appelstrasse 9a, 30167 Hanover, Germany; sester@ikg.uni-hannover.de
\* Correspondence: zourlidou@ikg.uni-hannover.de; Tel.: +49-511-762-19435

**Abstract:** Mapping with surveying equipment is a time-consuming and cost-intensive procedure that makes the frequent map updating unaffordable. In the last few years, much research has focused on eliminating such problems by counting on crowdsourced data, such as GPS traces. An important source of information in maps, especially under the consideration of forthcoming self-driving vehicles, is the traffic regulators. This information is largely lacking in maps like OpenstreetMap (OSM) and this article is motivated by this fact. The topic of this systematic literature review (SLR) is the detection and recognition of traffic regulators such as traffic lights (signals), stop-, yield-, priority-signs, right of way priority rules and turning restrictions at intersections, by leveraging non imagery crowdsourced data. More particularly, the aim of this study is (1) to identify the range of detected and recognised regulatory types by *crowdsensing* means, (2) to indicate the different classification techniques that can be used for these two tasks, (3) to assess the performance of different methods, as well as (4) to identify important aspects of the applicability of these methods. The two largest databases of peer-reviewed literature were used to locate relevant research studies and after different screening steps eleven articles were selected for review. Two major findings were concluded—(a) most regulator types can be identified with over 80% accuracy, even using heuristic-driven approaches and (b) under the current progress on the field, no study can be reproduced for comparative purposes nor can solely rely on open data sources due to lack of publicly available datasets and ground truth maps. Future research directions are highlighted as possible extensions of the reviewed studies.

**Keywords:** traffic regulators; traffic rules; traffic signs; VGI; crowdsourcing; SLR

---

## 1. Introduction

In recent years there has been an explosion of interest in creating and sharing geographic information by individuals that have been described as sensors—*citizens as sensors* who can capture various measurements at their local environments [1]. Since the establishment of smartphones and social media, this phenomenon has been growing only bigger and faster and the technology of an expensive work task being carried out by a group of people, known as *crowdsourcing*, is here to stay [2]. Interestingly, individuals contribute this information voluntarily, with most known volunteered geographic information (VGI) initiative the OpenStreetMap (OSM).

Guo et al. [3], discussing Mobile Crowd Sensing (MCS), distinguish two unique features—(1) it involves both implicit and explicit participation; (2) it collects data from two user-participant data sources—mobile social networks and mobile sensing. Either with minimum or major awareness and involvement, in a sensing system (opportunistic and participatory respectively) [4] volunteers can gather fast lots of geographic information that can be then processed for many purposes. Agamennoni et al. [5] and Zheng et al. [6] extract context information in form of *interesting* activities and places, Li et al. [7] mine road features such as road name and class and Niehöfer et al. [8] compute

velocity estimates per road segment for recommending routes of minimum predicted duration instead of shortest ones. Other popular topics based on crowdsourced data concern parking spot occupancy estimations [9], traffic state estimation [10], environmental noise monitoring [11,12], fuel-efficient map applications [13], and traffic condition sensing (bumps, potholes, hard braking, honking) based either on mobile devices [12] or on sensor-equipped vehicles [14], to name a few.

Some of the aforementioned approaches use data from various sensors to meet their purposes. Nevertheless, the minimum data needed for extracting some sort of geographic or location-aware information remains the time-ordered locations of a moving object. These locations are obtained as sequences of GPS recorded logs and compose spatiotemporal trajectories. Some examples of trajectories of different moving objects (pedestrian, bicycle, car, sailboat) are shown in Figure 1. Trajectories contain implicit knowledge of objects' movement regarding the underlying movement patterns and structure that can be identified with pattern recognition techniques and then can be applicable in various domains [15]. Traffic condition and patterns can be mined from trajectories [16,17], as well as intersection travel time [18]. Regarding movement patterns, a conceptual view of trajectories by identifying stops and moves [19] enable us to discover hidden patterns in object movement [20–22] or unveil interesting locations for individuals or group of people [23,24]. Other trajectory related applications concern trip purpose inference [25], sudden event detections such as braking incidents [26], road user behaviour classification [27], as well as anomaly (outlier) detection in various contexts, like traffic [28] or route navigation anomalies [29].

A field with much research interest lately is automatic map updating. According to Reference [30], roads change by as much as 15% a year. Map update refer both to the road network itself and the various features that come on top of the latter. Dozens of studies have been focused on the automatic generation of the road network from GPS tracks [31–35] and on subtopics that referred to the latter as main topic, such as intersection detection [36–39]. However, that interest is not uniform for the map feature categories that also need to be automatically updated. Such a relatively under-explored research field is that of traffic regulators.

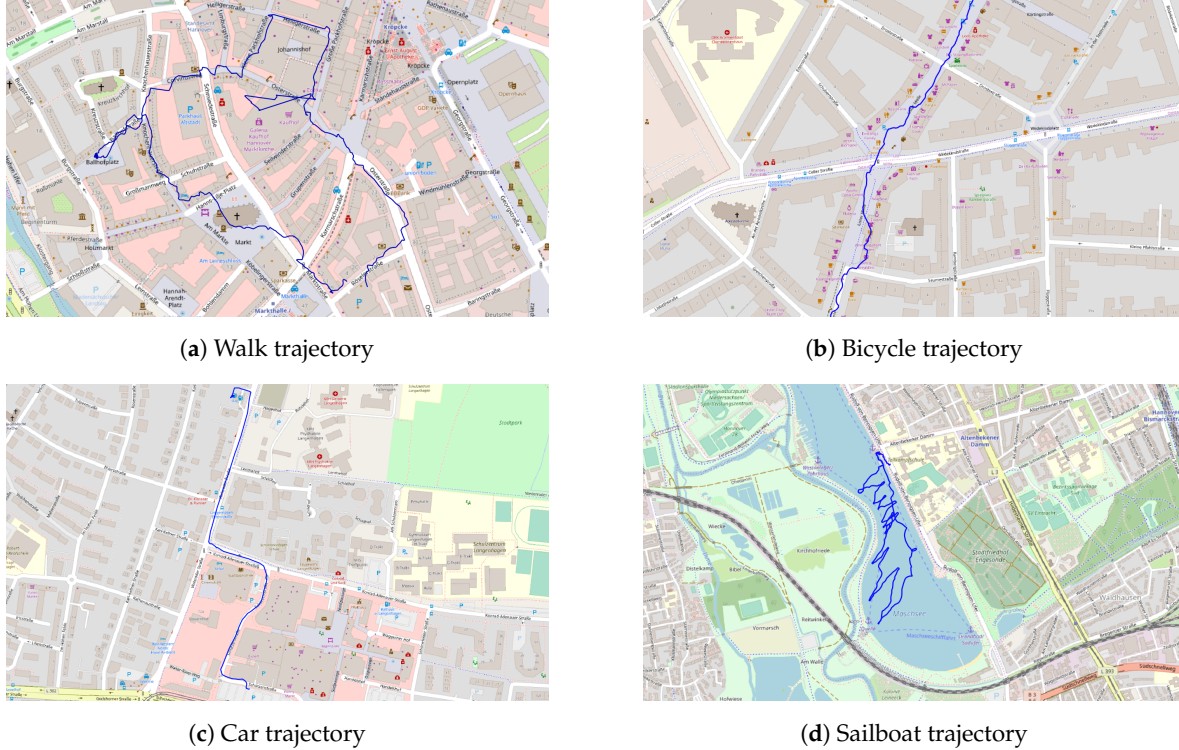

(**a**) Walk trajectory

(**b**) Bicycle trajectory

(**c**) Car trajectory

(**d**) Sailboat trajectory

**Figure 1.** Trajectories from different movement modes: (**a**) walking, (**b**) cycling, (**c**) driving and (**d**) sailing.

Road network consists of interconnected junctions which are geometrically complex locations. Such junctions are regulated with certain traffic regulators in sake of traffic participants' safety. By traffic regulators we distinguish two types: (a) in-situ traffic rules materialized as physical objects (traffic signs) and (b) global/general traffic rules such as the right of way priority. Traffic controls obviously affect the movement behaviour of the objects and might even affect the decisions that they take when selecting which navigation route to follow, such as avoiding complicated crossings [40].

Nevertheless, this information is yet largely missing from maps like OSM . This fact has motivated lately studies to explore how such information can be extracted automatically with crowdsourced means. Crowd-sourcing traffic regulation information has many benefits, as traffic rules are important components of maps considering the advances in self-driving vehicle field. Maps that contain such information can also contribute to driving safety [41] by assisting drivers to regulate their driving behaviour according to traffic rules through Advanced Driving Assistance Systems (ADAS). Such an example is the estimation of drivers' context awareness and issuing relevant warnings [42–45]. Moreover, fuel efficient route recommendations require such information for estimating fuel consumption and proposing routes accordingly [13]. Big location based service providers are also interested in including road signage information in their products. Such an example is HERE maps, which provides cloud-based service for delivering up-to-date traffic signage information to connected cars, so that drivers get warnings of changes (e.g., rerouting, traffic speed) along their routes [46]. Sensor event attributes required for that road sign service include among others latitude/longitude values, road sign type and road sign values captured by GPS and video camera devices.

Traffic regulator detection and recognition can be done mainly using data from two different sources—(a) images and (b) GPS tracks. Research on (a) involves traffic sign classification [47,48] using camera photos or mobile mapping systems as in Reference [49] where a traffic sign inventory is being built. Other computer-vision oriented methods try to predict the traffic signal phases [50,51] or timing [52], for facilitating the faster reach of destination. GPS tracks as a lightweight source of data that can be easily recorded with no special equipment (e.g., all smartphones nowadays have GPS receivers) and minimum user involvement in the recording task (no special need for placement a camera for capturing images) can be seen as a more *friendly* traffic-regulator crowd-sensing approach and exactly that is the topic of the SLR presented in this article. A visual depiction of the problem reviewed here, is shown in Figure 2 and can be shortly summarized in the research question *How from lightweight crowdsourced data can traffic regulators be detected and categorized?*

To our knowledge, no review of studies related to traffic regulator detection based on non-imagery data has been published so far. Given the strong interest in, and practical demand for, up-to-date maps, this work aims at locating all relevant studies, extracting important aspects of their proposed methods, assessing their performance and identifying possible research directions that can lead to solutions that can *massively* and *practically* enhance maps with traffic regulators.

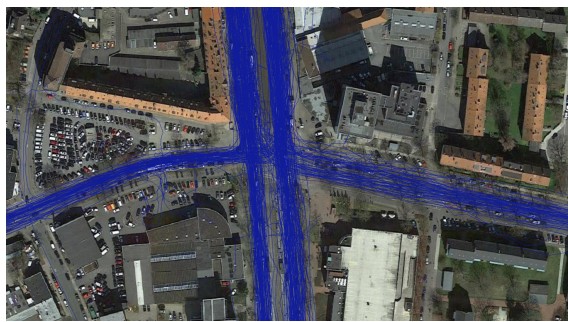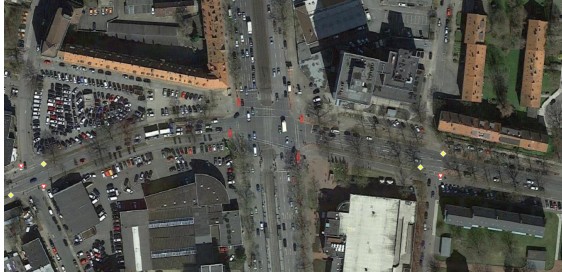

(**a**) Vehicle trajectories crossing an intersection.　　　　(**b**) A traffic-signal controlled junction.

**Figure 2.** From crowdsourced data (**a**) to traffic regulator detection and recognition (**b**).

In the following sections, we describe the methodology we used to conduct the *systematic* literature review (Section 2) and we present the results of the SLR in Section 3. A discussion over the findings is given in Section 4.

## 2. Materials and Methods

Although the method of SLR was originally developed within the medicine research field, lately it has been adopted from various GIS studies [53–56]. Torraco [57] explains that an "integrative literature review is a form of research that reviews, critiques, and synthesizes representative literature on a topic in an integrated way such that new frameworks and perspectives on the topic are generated," which summarizes the motivation of this SLR. A SLR is a kind of review that needs to be organised around specific research questions or purposes and have a structure that allows the researcher to develop criteria for including or excluding research publications from the final synthesis. Such criteria are determined according to the focus of review which might be research outcomes, methods, theories, or practices or applications [58]. The focus of this SLR are on the methods of detecting traffic regulators.

Moreover, a SLR makes use of detailed, rigorous and explicit methods [59]. According to Xiao and Watson [60], although different types of literature reviews can be implemented following various procedures, all of them in general can be conducted following eight common steps: (1) formulating the research problem; (2) developing and validating the review protocol; (3) searching the literature; (4) screening for inclusion; (5) assessing quality; (6) extracting data; (7) analysing and synthesizing data; and (8) reporting the findings. Equally important elements of a SLR are its validity, reliability and repeatability [60] and for this reason special attention should be paid on the methodology of conducting and documenting it.

Adopting this 8-step process, our research problem was formulated in four questions as shown in Table 1. Mainly, we want to (1) identify what kind of regulators can be detected and recognised by processing crowdsourced non-image data (*RQ1*); (2) summarize what kind of methods are being used for the scope of 1. (*RQ2*); (3) assess how well those methods perform on predicting the regulator category (*RQ3*); and last (4) recognise probable under-explored aspects of the problem that through this work can become visible to scientific community for further exploration (*RQ4*).

After that, a protocol was defined according to Prisma 2009 checklist [61] and the literature search was conducted as described in Section 2.1. The screening and eligibility steps are explained in Section 2.2. We selected to integrate each of individual studies semi-quantitatively [59] that, contrary to qualitative ones, combine the results of reviewed studies statistically. That way they provide a comparative insight of how methods differ. We call our method semi-quantitative, as in some cases a statistical analysis was not possible, as explained in Section 3. We made a selection of data to be extracted for the semi-quantitative assessment (Table 6) and the results of the analysis and synthesis are reported in Section 3.

**Table 1.** The Systematic Literature Review (SLR) objectives formulated as research questions.

| SLR Research Questions | |
|---|---|
| *What* | What traffic regulators have been detected and recognised based on crowdsourced non-image data? |
| *How* | What methods have been used for this purpose? Under what settings the methods have been tested? (dataset, participants, ground truth map )? |
| *How well* | What is the performance of these methods? |
| *Other* | Are there any under-explored aspects of the topic? |

*2.1. Literature Search*

2.1.1. Channels for Literature Search

There are three main sources for obtaining a complete list of literature [60]—(1) electronic databases; (2) backward searching; and (3) forward searching. We searched these three sources as following. We selected two of the biggest multidisciplinary scientific citation and abstract indexing databases (Table 2), namely *Scopus* [62] and Web of Science [63], each having more than 69 millions records. Having created an account in these two citations services, the search queries and the retrieved list of documents were saved and exported in separate files as part of the documentation that the SLR protocol calls for [61]. As explained later in Section 2.1.2, these databases provide *advanced search* of documents based on personalised criteria. Therefore before using such advanced search to query articles based on certain criteria, special attention was given on carefully reading the search instructions and the various examples provided by the two service carriers.

The number of articles at each step of the SLR *Screening* process are indicated in the *Step* columns of Table 2. *Step 1* indicates the initial number of articles that the search engines returned after the search string was given. At *Step 2* the articles from the previous step are screened based on their *Title*. Similarly, at *Step 3, Step 4*, articles are screened according to their Abstract and Full-text respectively. Articles with the same topic from the same author(s) are screened at the *Step 5*, where the latest and usually most advanced article is selected for the final list of the under review articles.

**Table 2.** Electronic databases and search engines used in the SLR.

| Source | URL | Step 1 | Step 2 (Title) | Step 3 (Abstract) | Step 4 (Full-text) | Step 5 (Author) |
|---|---|---|---|---|---|---|
| Scopus | https://www.scopus.com/ | 603 | 171 | 36 | 8 | 6 |
| Web of Science | https://apps.webofknowledge.com/ | 326 | 57 | 9 | 6 | 4 |
| **Total** | | 929 | 228 | 45 | 14 | 10 |

The Backward search, known also as *snowballing* [64], involved identifying relevant work cited by articles, making use of the list of references placed at the end of an article. The Forward search was also used for the same purpose by locating articles that have since cited the articles reviewed. We used Google Scholar for conducting the Forward search. The process of Backward and Forward search is explained in detail in Section 2.2.3.

Last, as an additional channel of literature we used our personal knowledge, that is, according to Reference [64], related to what we already knew and who we knew, for example, our existing knowledge and resources, our personal academic contacts and networks, as well as serendipitous discovery, such as locating a relevant paper when looking for something else. We refer to this channel of literature as *other sources* in the rest of the article.

2.1.2. Query Terms

The selection of the terms for querying articles is of high importance as it significantly affects the number of articles that are retrieved and determines their relevance to the topic. Neither an enormous amount of found articles nor a small one is a good starting point of an SLR in the first case the screening process may take way too long time to be accomplished within a reasonable time-frame and human resources and in the second case, the review may lack of sufficient source of input for reaching a quality level that ensures a broad and complete enough assessment of the current state of the literature. This means that query terms can both eliminate and broaden the results of the search and, for this reason, they should be carefully selected.

For selecting the set of query terms we used the following strategy. First we created a list of all possible terms (also synonyms) that corresponds to the concept of *traffic rules*, recording also all the possible forms (grammatic and spelling) within the context of a phrase (e.g., plural forms,

combined words separated with hyphen like *traffic-signal*, etc). Then, by using the advanced search of the two databases and each term separately, we examined the number of articles and the first pages of the retrieved list, for assessing the quantitative and qualitative result of the search. Moreover, we experimented with searching terms within only the text of the title, within the abstract and then in combinations of title-abstract-keywords. Thereby, we found out that since the number of articles related to the term *traffic signals* was enormous (some thousands), we had to restrict the findings to those that include both the terms *traffic sign* and *GPS*, either in the title text or in the abstract or in the keyword part of the articles. For the other terms we eliminated the search only within the title text. That way, both the number of found articles and their content was matched better to our requirements.

For implementing the aforementioned strategy we used the logical operators "OR" and "AND" for combining different search queries and the asterisk "*" for retrieving words with variant zero to many characters. These operators have similar function at both databases. Double quotations though work differently (loose phrases in Scopus and exact in Web of Science). Also in Scopus, punctuation is ignored (hyphen is treated as punctuation and therefore ignored if it is not in exact phrases) and plural and spelling variants are included (*traffic sign* includes *traffic signs*). Web of Science by default uses lemmatization, which means it "makes use of dictionaries that define pairs and clusters (e.g., defense, defence) of words with the same meaning or with a shared morphological structure" [63]. Synonyms and lemmatized terms are turned off when quotation marks are used or wildcards. In Table 3 are shown the search terms that were used for article retrieval. The final search string is shown in Table 4.

**Table 3.** Query terms that were used for searching the Web of Science and Scopus for the two SLC concerning concepts.

| Query Terms | |
| --- | --- |
| traffic regulator * | traffic rule * |
| traffic restriction * | turning restriction * |
| crowdsourc * traffic | crowdsens * traffic |
| intersection classification | intersection regulation |
| intersection control * | intersection regulat * |
| junction control * | junction regulat * |
| junction regulat * | junction classification |
| traffic sign * | GPS |

**Table 4.** The Query string that was used in Scopus search engine.

| Search Query String |
| --- |
| TITLE ("traffic regulator *" OR ("traffic rule *") OR ("traffic restriction *") OR ("turning restriction *") OR ("crowdsourc * traffic") OR ("crowdsens * traffic") OR ("intersection classification") OR ("intersection regulation") OR ("intersection control *") OR ("intersection regulat *") OR ("junction control *") OR ("junction regulat *") OR ("junction regulat *") OR ("junction classification") ) OR TITLE-ABS-KEY ("traffic sign *" AND "GPS") |

Last, three extra filters were applied. The range of the publication year was set from 1999 to 2019, the language of the documents to English and the subject category (*subject area* in Scopus and *Web of Science Categories caterories*) by excluding the irrelevant ones (e.g., medicine, biology, social sciences, etc).

*2.2. Screening and Eligibility Check*

Having as input all the articles that were retrieved from the two databases, the next step of the SLR was the screening and eligibility check, according to explicitly defined criteria. First the articles were eliminated based on their title. All the articles that referred roughly to computer vision methods or whose scope was related to accidents, regulation violation detection, driving intention prediction, localization or formalisation of traffic rules in the context of autonomous vehicles, were excluded. That

way the number of articles that passed to the next stage was small enough so that their abstracts could be read within reasonable time. The articles that were eligible according to the inclusion criteria were passed to the next step, where they had to be fully read. The same articles retrieved from different databases were read once. After this step, a list of papers came up that was enriched with an article found from other sources (Section 2.1.1). We ended up with a pre-final review list that we searched Backward and Forward for relevant articles. That step was very important as we spotted studies that we had not located with the previous time-consuming process. The resulting articles made up the final review list.

### 2.2.1. Inclusion and Exclusion Criteria

As mentioned in the previous paragraph (Section 2.2), the elimination of the articles was done based on criteria that were defined along with the research questions. By having formulated the scope of the SLR in very specific research questions, determining the inclusion and exclusion criteria ended up to be straightforward. These criteria are enlisted in Table 5.

We regarded three mainlines for including a study in the review list—(1) the first is related with the study object itself (*What* from RQ1) and with the method employed for pursuing that objective (*How* from RQ2). Studies not directly related with the topic were excluded, such as References [65,66], which focus more on traffic regulation violation detection. Reference [67] was also excluded as it focuses on enriching maps for traffic sign compliance. We found some other articles with objectives close to the SLR topic, such as References [18,27,68], but since their topic did not really coincide with the latter, they were also excluded. (2) The objective, the methodology as well as results of the tested method should be clearly given in the the body of the article. There were cases of articles talking about the usage of a certain method for detecting traffic regulations without providing evidence from quantitative results coming from experiments and testing, for example, References [22,41,69]. These articles were not included due to invalidity of the *Incl2* criterion. (3) Last, only the more advanced article from the same authors on a certain topic was included in the review list, while earlier and more often shorter versions were excluded [70–73]. That way we eliminated duplicate studies.

**Table 5.** The defined inclusion and exclusion criteria that were applied during the Screening and Eligibility steps of the SLR.

| | |
|---|---|
| *Incl1* | The article is relevant with the detection and recognition of traffic regulators using non imagery crowdsourced gps tracks. |
| *Incl2* | The article describes in a clear way the objectives, methods and results of the research. |
| *Incl3* | The article is the most recent work of an author that has written earlier a short paper or a similar article on exactly the same topic, decribing the same method. |
| *Excl1* | The article refers to a computer-vision based traffic-regulator detection approach, e.g., detection based on camera or satellite image. |
| *Excl2* | The article lacks of clarity in depicting its objectives, methods and results or\and it mentions the applicability of their method on SLR's objective but it does not report any results on that. |
| *Excl3* | Authors have published a more detailed or advanced article on the same topic with the same objective and method. |

### 2.2.2. Applying the Inclusion and Exclusion Criteria

The decision for including or excluding an article came after three rounds of scanning, where for each the validity of the inclusion-exclusion criteria was checked and articles were either passed for a further scanning or definitely excluded from the review process. These distinct processing steps are depicted in Figure 3. With title screening, basically articles that possibly matched the *Incl1* were passed to the next phase. Since criteria *Incl2* (*Excl2*) and *Incl3* (*Excl3*) regard the methodology and result of the proposed approaches, they were not expected to filter out articles at this stage. Those successfully screened by title were assessed based on *Incl1*(*Excl1*) and *Incl2*(*Excl2*) by their abstract and unless not excluded, by their full text. Until this point, the articles' eligibility was validated for *Incl1* (*Excl1*) and *Incl2* (*Excl2*) criteria. Duplicates were eliminated according to *Incl3* (*Excl3*) and the

review list of articles come from the first channel of literature search was finalised. This list was then promoted for Backward-Forward search.

### 2.2.3. Snowballing Search and Personal Knowledge Enrichment

The article list from the previous screening step (Figure 3) was first enriched with articles from our personal knowledge and then all of them were searched Backward and Forward. Here there is an iteration of the screening steps explained before, for the articles cited by the selected ones (backward) or from later published studies that cite the aforementioned (forward). The final review list was finally compiled containing eligible articles from all channels of literature, as presented earlier in Section 2.1.1.

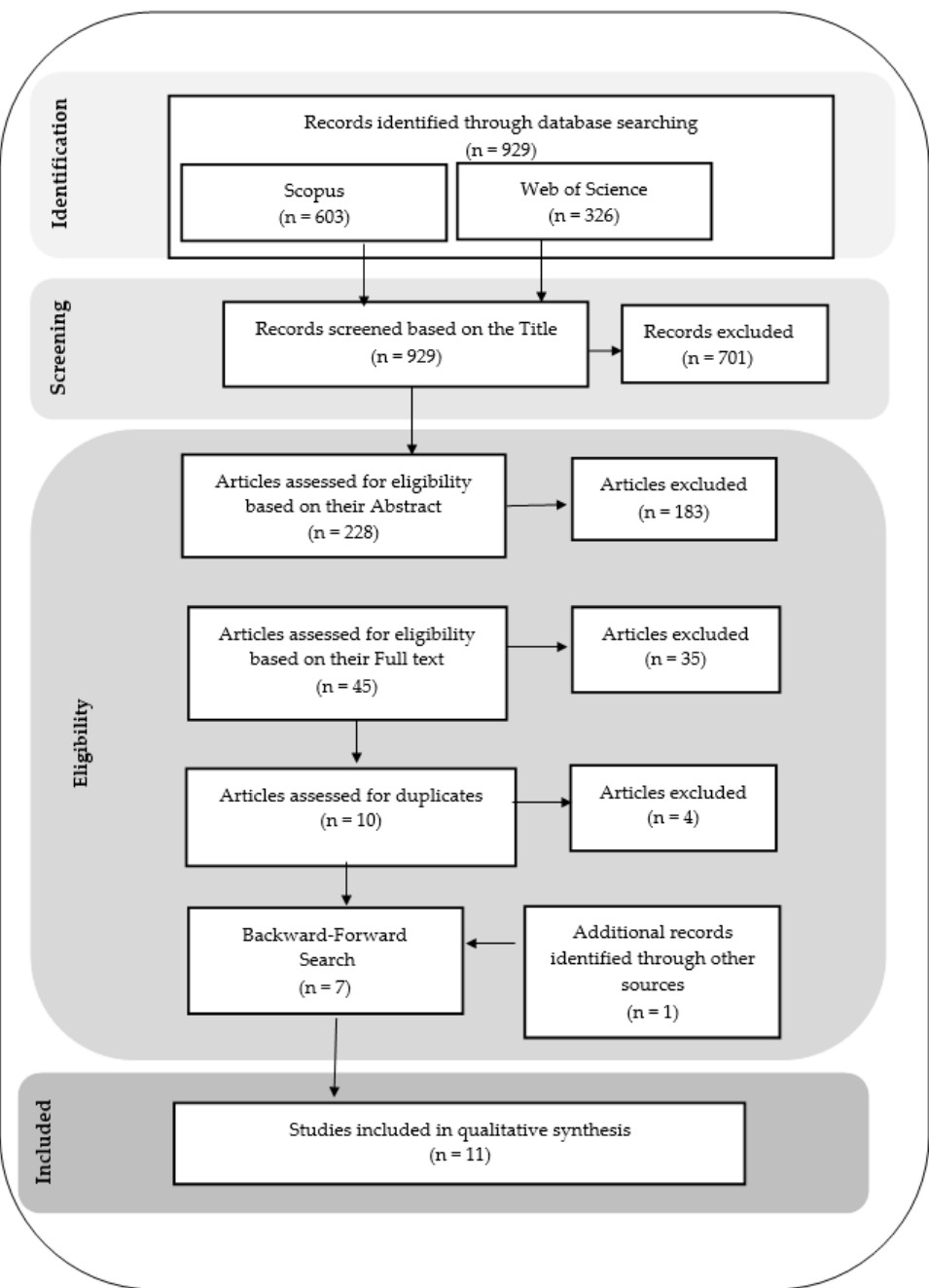

**Figure 3.** PRISMA Flow diagram [61] of the literature search and articles' selection for the SLR.

## 2.3. Information Extraction

According to Reference [57], *the best literature reviews examine the literature with a particular lens defined by the article's objectives.* This means that they rarely examine all aspects of previous research but rather this lens points the author and reader to certain aspects of studies that indeed are critically examined and evaluated. In this SLR the aspects to be further analysed came up after examining the research questions and the problem of classification itself. We are interested in identifying how a classifier can be built for regulator classification, how well it can predict different categories, how sufficiently it can perform on an other than the training geographic context, how many samples are needed for learning the various labels, what kind of data can be used for the scope, and all these questions we would like to answer under the prism of identifying methods or characteristics that can be reproduced in order to *practically* enrich maps with that important information. Ultimately, we want to answer (a) whether someone with today available tools and data could detect regulators and, if yes, which of them and under what limitations, (b) what the "recipe" would be for that, and (c) what still remains unexplored or under-explored. The aspects that are analysed in this SLR are enlisted in Table 6 and are explained in more detail in the next paragraphs.

**Table 6.** Comparison aspects (dimensions) of the reviewed articles in terms of their proposed methods, experimental settings and performance.

| Dimensions | Description |
|---|---|
| Regulator category | Regulators for which a detection methodology is proposed |
| Number of regulators | Diversion of regulator categories |
| Study area | Country, City |
| Participants | Number of participants whose data are crowdsourced |
| Dataset type | Publicly available dataset or not |
| Dataset size | Size of crowdsourced data |
| Dataset timespan | Time range of the dataset used in the study |
| Ground truth map source | Acquisition method of ground truth map (on-site inspection, official public or non-public provider) |
| Cross-city testing | Results from cross-city testing are provided in the study or not |
| Minimum number of samples | Minimum number of crossings per junction needed for sufficient classification performance |
| Classification method | Classification methods that were used for learning to detect and recognise regulators |
| Classification features | Features used for classification |
| Classification Accuracy | Classifier's accuracy (best performance) |

*Regulator (Categories and Diversion)*—Ideally a method could identify all different regulator categories, but in most of the cases each method focuses on a certain subset of traffic rules, mainly as discussed later due to limitations imposed by dataset availability. For each article we identified the traffic rules that the proposed approach relates to.

*Area of Study*—In the GIS field context, often research takes place in or concerns different geographic locations for studying the effect of location on the examined variable. In the context of this research, although traffic regulations have the same meaning no matter the geographic context, we extracted the country and the city(cities) that the study deals with, so that we can identify whether the same method (if any) can perform equally well in different geographic context.

*Participants*—The focus of this SLR is limited to crowdsourced non-image data and the motivation for that comes from the advantages that crowdsourcing in general offers. As Heipke [2] points out, the field is constantly growing and has lots of potential for mapping, mainly due to two reasons: (a) the technology is mature enough to stay for long, (b) other means of mapping are too slow or too expensive. We exported the numbers of participants whose data were recorded and contributed to study's dataset and varies from 1 to several individuals.

*Dataset Type*—The type of the dataset can be publicly available (e.g., an open dataset) or not. Here we denote as dataset both GPS tracks (or non GPS data exported from OSM) and ground truth maps.

Therefore, we categorized datasets into publicly and non-publicly available. We regard this piece of information as important, as we want to find out whether today someone can download a dataset, train and test the method on that and then use the classifier on newly crowdsourced data without the need to discover the ground truth map for the new data, which is in general a time-consuming process.

*Dataset Size*—Dataset size information is extracted in form of number of trips (trajectories) and number of junctions sampled from the traces. This information shows how extended the study is and sheds light on the generalisations that can be drawn up from the results.

*Dataset Timespan*—The timespan of the dataset is important to assess seasonal limitations and examine possible extensions of the methods regarding the repetition of the detection process.

*Ground truth Map Acquisition Method*—We exported the acquisition method of ground truth map. If stated in the article, we noted the source or the method that has been used, else we noted it as non-stated. Without knowing the ground truth map neither classifiers can be trained nor classification results can be validated from any selected classification method, either involving training or not.

*Cross-City Testing*—It is important to know whether a classifier trained and tested on city A can equally perform on city B. Not all studies explore this aspect of the problem, so we categorise them according to it.

*Number of passes*—This information is related obviously with the dataset size, but here the motivation is to identify the minimum amount of crossings per junction needed for classifier's optimization. We exported the minimum number of samples documented by authors if they conducted such experiments, else we noted this piece of information as non-explored.

*Classification Method*—The type of classification methods are exported from the studies to find out how different is one approach from the other (if they use different classifiers) and spot those that perform better than others.

*Classification Features*—Physical values as well as statistical ones are used as features that feed classifiers. Those measures were exported to examine the variety of features that have been used so far and to discover possible combinations or new features that have not been tested so far.

*Accuracy*—The accuracy of predicted regulators was also exported for evaluating the progress on the field. Since accuracy is a measure of how sufficiently a method can categorize junctions according to regulators, we used this information to find out possible directions that need further exploration for improving the current results.

## 3. Results

As was explained in Section 2, we searched the two literature databases leading us to locate, in total, 929 articles (Figure 3). These articles were first screened based on their title. In total, 701 of them were excluded as not relevant to the topic and 228 were passed to the next eligibility screening. The abstracts of those 228 articles were read and only 45 were judged eligible for full-text analysis. From a full-text reading only 10 were classified as eligible for SLR. Four articles out of 10 were excluded as they were identified as duplicates (Table 7). In the 6 remaining articles, 1 article was added from other sources and the 7 articles were searched Backward and Forward. The latter search identified 4 more articles. Therefore, 11 papers complete the final review list, which are listed in Table 8 in chronological order.

**Table 7.** Eligibility processing steps after full-text reading.

| Eligibility Processing Steps after Full-Text Reading | Articles |
| --- | --- |
| Articles after full-text reading | 10 |
| After duplicate exclusion | 6 |
| After adding from other sources | 7 |
| After Backward-Forward Search | 11 |
| **Final** | 11 |

**Table 8.** Articles selected after the *Screening* step of the SLR process in chronological order.

| a/a | Reference | Author(s) | Title of the Article | Type | Year | Classified Regulator |
|---|---|---|---|---|---|---|
| 1 | [74] | Pribe & Rogers | Learning To Associate Observed Driver Behavior with Traffic Controls | Journal | 1999 | Stop-Signs, Traffic-Signals |
| 2 | [52] | Carisi et al. | Enhancing in Vehicle Digital Maps via GPS Crowdsourcing | Conference | 2011 | Stop-Signs,Traffic-Signals |
| 3 | [75] | Hu et al. | SmartRoad: Smartphone-Based Crowd Sensing for Traffic Regulator Detection and Identification | Journal | 2015 | Stop-Signs, Traffic-Signals, Uncontrolled |
| 4 | [76] | Seremi & Abdelzahe | Combining Map-Based Inference and Crowd-Sensing for Detecting Traffic Regulators | Conference | 2015 | Stops, Traffic-Signals |
| 5 | [77] | Aly et al. | Automatic Rich Map Semantics Identification Through Smartphone-Based Crowd-Sensing | Journal | 2017 | Stops, Traffic-Signals |
| 6 | [78] | Efentakis et al. | Crowdsourcing Turning Restrictions from Map-matched Trajectories | Journal | 2017 | Turning restrictions |
| 7 | [79] | Wang Chao et al. | Automatic Intersection and Traffic Rule Detection by Mining Motor Vehicle GPS Trajectories | Journal | 2017 | Traffic-Signals |
| 8 | [80] | Méneroux et al. | Detection and Localization of Traffic Signals with GPS Floating Car Data and Random Forest | Conference | 2018 | Traffic-Signals |
| 9 | [81] | Munoz-Organero et al. | Automatic Detection of Traffic Lights, Street Crossings and Urban Roundabouts Combining Outlier Detection and Deep Learning Classification Techniques Based on GPS Traces while Driving | Journal | 2018 | Traffic-Signals, Street Crossings, Roundabouts |
| 10 | [82] | Qiu et al. | Towards Robust Vehicular Context Sensing | Journal | 2018 | Stop-Signs |
| 11 | [83] | Zourlidou et al. | Classification of Street Junctions According to Traffic Regulators | Conference | 2019 | Traffic-Signals, Yied/Priority Junctions, Uncontrolled |

We analysed the articles based on year of publication as can be seen in Figure 4. We made two observations—(1) after the first article published on the topic (1999), there is a long *inactivity* period and only after 2011 we can see that the topic seems to gain again the interest of the scientific community. After 2015, the interest started increasing and given this study was conducted in the middle of 2019, no conclusions can be done for the ongoing year.

Moreover, we examined the scientific disciplines that are involved in the studies based on the journal category. More specifically, we analysed the articles found from Web of Science, after the abstract eligibility screening. As can be seen from Figure 5, most of the articles are from the Computer Science discipline and interestingly no paper was published in the Geoinformation or Transportation Science thematic category journal. This is a contradictory finding to our expectation as the topic itself is, if not *pure*, at least geoinformation related.

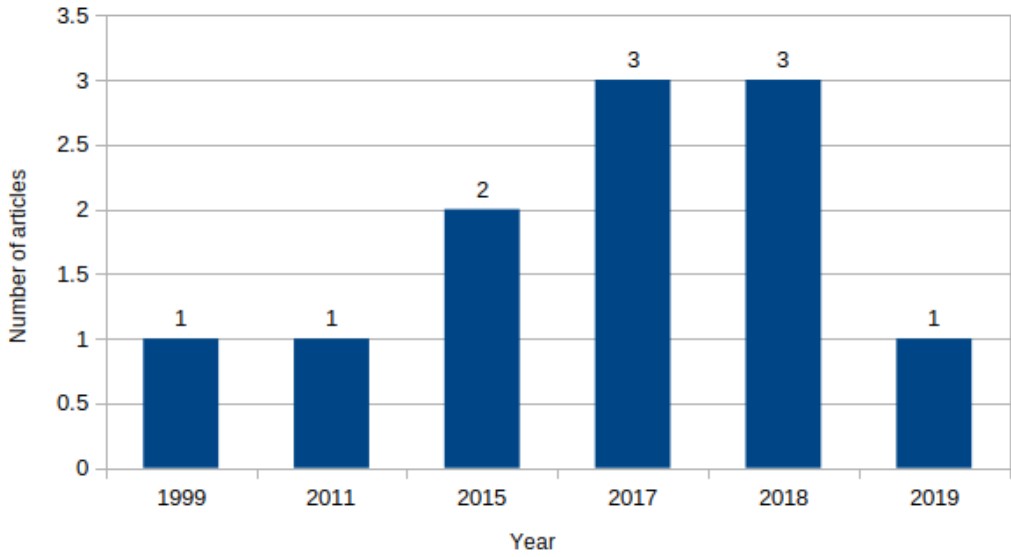

**Figure 4.** Number of SLR articles per year of publication.

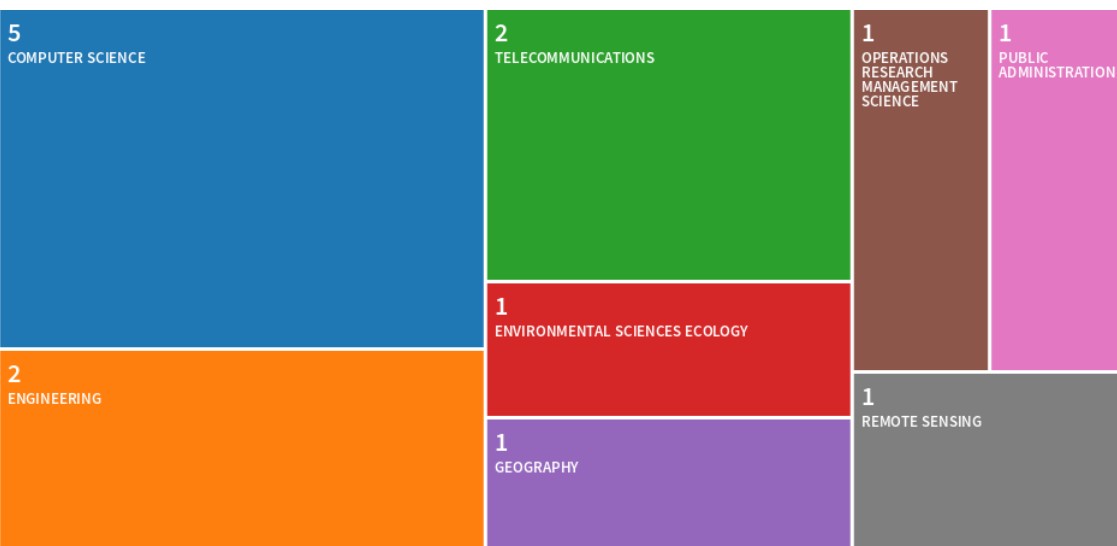

**Figure 5.** Treemap visualisation of categories of Web of Science found articles, after full-text screening.

The fact that four out of eleven articles were spotted with the *snowballing* method and the fact that most of the articles are published in Computer Science journals, motivated us to analyse further

the found articles and more specifically their titles. We applied a text analysis to their titles as shown in Table 9, for identifying the common terms that are used in the title text. This analysis explained why those five articles weren't spotted directly from the database search. As the list with the frequent terms indicates, keywords like "regulator" met only twice in the whole title of the eleven articles and others like "signal"(s), "light"(s), "control"(s), "junction"(s) and "intersection"(s) only once or none. Going back to the titles themselves, we can see that articles such as References [74] and [52] have titles of quite *broad* meaning, making hard their association with the keyword search (*Learning to associate observed driver behaviour with traffic controls* and *Enhancing in vehicle digital maps via GPS crowdsourcing*). Regarding the journal category, no doubt the methods involved in regulator detection and recognition originate from computer science discipline (classification, clustering, etc.) and maybe the broader-meaning titles explain why most of the articles were not published in geoinformation related journals. These observations emphasise further the objectives of this SLR that, among others, is, as stated earlier, to *reintroduce* the topic to the geoinformation society or at least to make the problems that the topic involves (more) visible to the geoinformation community. In the following paragraphs, the results of information extraction according to the identified problem aspects (Table 6) are discussed.

**Table 9.** Occurrences and frequency of the terms that are found in title text (all title as a single text) of the SLR selected articles (shown for occurrences greater than two).

| Word | Occurrences | Frequency | Rank |
|---|---|---|---|
| traffic | 7 | 6.9% | 1 |
| sensing | 5 | 4.9% | 2 |
| detection | 5 | 4.9% | 2 |
| gps | 4 | 3.9% | 3 |
| based | 4 | 3.9% | 3 |
| automatic | 3 | 2.9% | 4 |
| map | 3 | 2.9% | 4 |
| crowd | 3 | 2.9% | 4 |
| vehicle | 2 | 2% | 5 |
| identification | 2 | 2% | 5 |
| regulators | 2 | 2% | 5 |
| street | 2 | 2% | 5 |
| trajectories | 2 | 2% | 5 |
| learning | 2 | 2% | 5 |
| smartphone | 2 | 2% | 5 |
| combining | 2 | 2% | 5 |
| classification | 2 | 2% | 5 |
| crowdsourcing | 2 | 2% | 5 |

### 3.1. Regulators: Categories and Diversion

We identified six different regulator categories that are detected with crowdsensing means—traffic lights (TL), stop signs (SS), right of way rules (RW) for uncontrolled intersections, turning restrictions (TR), priority sign (PS) and yield sign (YS). The regulator category rate across the eleven studies is depicted in Figure 6a. TL is the more *popular* regulator within the studies, with 40% percentage, and SS (30%) and RW (15%) follow. TR is detected only by 10% of studies and less popular regulators are PS/YS (5%). Here we should note that some studies, except for those on traffic regulators, detect other map/street elements, such as street crossings and roundabouts in Reference [81] or underpasses, stairs, escalators, footbridges, crosswalks, elevators, ramps, and so forth, as described in Reference [77]. Since this SLR focuses on intersection controlling categories, we ignored these elements from the study. Nevertheless, we report in the diversity field in Table 10 in parentheses the total detected subjects. Similarly, the two studies that detect TR examine subcategories of turning restrictions (e.g., right, left, straight, U-turn). We denote those subcategories also in parentheses in the diversity field of Table 10.

The diversity of the rules examined within each study was also analysed and the results are shown in Figure 6b. By diversity, we mean the number of different regulators that are detected within the same study. As mentioned earlier, ultimately a study should categorize as many regulators as possible. The maximum diversity is three and the minimum is one. No study found having diversity equalled four or five.

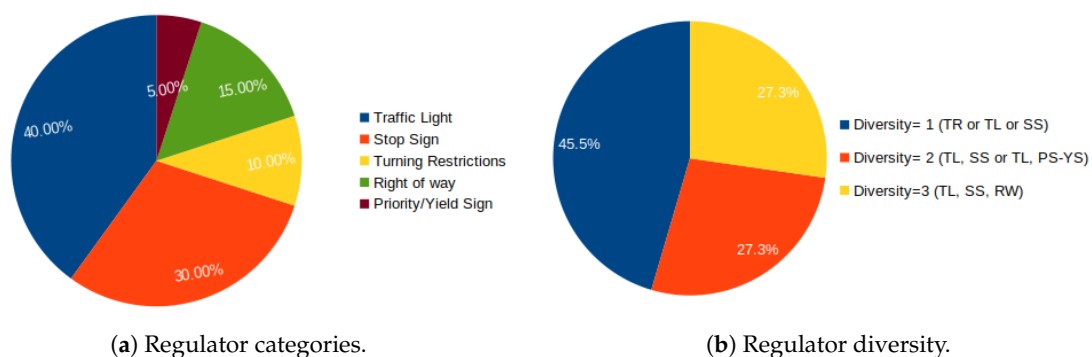

(**a**) Regulator categories.      (**b**) Regulator diversity.

**Figure 6.** Regulator categories and diversity across SLR reviewed studies.

### 3.2. Area of Study

One third (33%) of the experiments of the studies used datasets that are sampled in the USA (US) and 20% in Germany (DE), as can be seen in Figure 7a. One article may have many areas of study. In terms of area diversity within the same study, as Figure 7b suggests, the maximum is three and concerns 9.1% of the studies and the minimum is one concerning the majority of studies (72.7%).

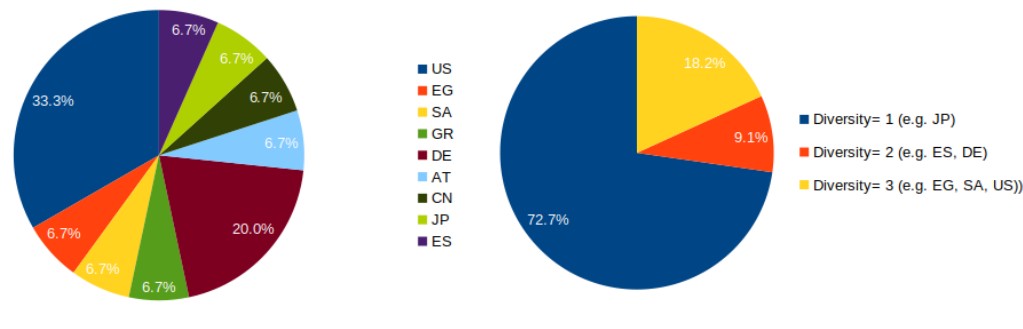

(**a**) Distribution of areas of studies (countries).      (**b**) Diversity of areas of study.

**Figure 7.** Diversity of areas of studies within the same study.

**Table 10.** Extracted information from the reviewed articles.

| Ref. | Reg. Category* | Reg. Diversity | Study Area▽ | Particip. | Open Dataset | Dataset Size | Dataset Timespan | Ground truth Map Source | Cross-city Test. | Min Samples | Classif. Method | Classif. Features | Classif. Perform.◇ |
|---|---|---|---|---|---|---|---|---|---|---|---|---|---|
| 1. [74] | TL, SS, RW | 3 | US | - | No | 50 Reg. | - | Various | No | No | Neural Nets | Statist. | 100% |
| 2. [52] | TL, SS | 2 | US | - | Both | 32 Inter. | 3 Days | On site | Yes | Yes | Heuristics | Slowdowns, Standstills | >90% |
| 3. [75] | TL, SS, RW | 3 | US | 35 | No | 463 Reg. | 2 Mon. | On site | No | No | Random Forest, Spectral Clust. | Statist. | >90% |
| 4. [76] | TL, SS, RW | 3 | US | 46 | Both | >1K Inter. | - | Google Street Views | Yes | No | Random Forest | OSM, Statist. | >95% |
| 5. [77] | TL, SS | 2(9) | EG,SA,US | 5 | No | 24 km foot | - | On site | No | No | Heuristics | Dwell. Dur. | Optimal Prec., Rec.: 0.8 |
| 6. [78] | TR | 1(4) | GR, DE, AT | >2K | No | Mi./Bi. Turns | 1 Year | Various | Yes | No | Heuristics | %Turns | 66–77% |
| 7. [79] | TR | 1(12) | CN | 5 | No | 285 Inters. | - | On site | No | No | Clustering | Headings, Time-Series Points | - |
| 8. [80] | TL | 2 | JP | - | No | 253 TL Inter. | 1 Mon. | On site | No | No | Random Forest | Stop Dur. | >85% |
| 9. [81] | TL | 1(3) | ES, DE | 1/10 | Both | 8.1 km, 55 Traj/ 23.6 km, 20 TL | - | On site | No | No | Deep Belief Net. | Speed, Accel. | Rec.: 0.89, Prec.: 0.88 |
| 10. [82] | SS | 1(4) | - | 6 | No | 55 Int. | 9 Mon. | On site | No | No | Heurisitcs | Stop patterns | Rec.: 0.86, Prec.: 0.90 |
| 11. [83] | TL, PS-YS | 2 | DE | - | No | 31 Inter. | - | On site | No | No | C4.5 | Speed Seq. | Rec.: 0.83, Prec.: 0.31, F-score: 0.45 |

* TL: Traffic-Lights, SS: Stops-Sign, RW: Right Way Rule, TR: Turning-Restrictions, PS: Priority-Sign, YS: Yield-Sign; ▽ Country names: US (USA), DE (Germany), GR (Greece), AT (Austria), CN (China), EG (Egypt), SA (Saudi Arabia), JP (Japan), ES (Spain); ◇ Prec.: Precision, Rec.: Recall.

### 3.3. Participants

The number of participants that collect the data is very diverse, from 1 subject to more than 2000. Also, four out of eleven studies do not report this parameter.

### 3.4. Dataset Type

Of the studies, 72.73% were conducted using datasets that are not publicly available. The other 27.27% used both publicly and non-publicly datasets. In the latter case, all but one used data acquired from OSM and only one study [81] reports the (non OSM) data source, which one can use to download data from. Obviously, even using open data sources like OSM, unless details are given regarding the data acquisition, one cannot get access to the exact dataset that other studies had previously used. In one case [76] where data was used from both sources, features were derived from OSM and combined in classifier's feature vector along with features computed from own collected data. In another case [52], open data were used only for testing algorithms but without verifying their results, most probably due to ground truth map unavailability. Last, the third study that uses both types of datasets [81], applied the proposed methods separately to each dataset, since for the public dataset the ground truth map was also provided.

### 3.5. Dataset Size

The size of the data used in the studies is very diverse as well as diverse in the way that it is reported in the articles, as can be seen in Table 10. Sometimes it is reported as the number of intersections, other times as the number of intersection approaches, regulators, length of trajectories, number of trajectories, recording time, and so forth. In most of the cases not all the aspects that describe precisely a dataset are given (e.g., number of sampled intersections is given, but not the number of trajectories that sample the area of interest). Here we should note that due to limitation of space, in Table 10 only some of the dataset size description aspects are reported that are given in the articles. Another observation is that more than half of the studies are small-scale according to the size of the dataset that they use (less than 100 intersections or regulations). The rest of the studies are again diverse—they use more than 190 intersections, with one study using over 1000 junctions and another one reporting billions of turn instances at a non-stated number of junctions [73].

### 3.6. Dataset Timespan

The timespan of data collection in seven out of eleven studies is not reported. From those that report it, we can deduce that the bigger the timespan is, the bigger the acquired dataset is.

### 3.7. Ground Truth Map Acquisition Method

The method for acquiring the ground truth map (Figure 8b) in 63.6% of the studies is on-site or direct observation and 18.2% of the articles use multiple sources for ground truth verification which may include direct observation and information from the transportation official agency, such as in Reference [74] or information from satellite images and on site examination, such as in Reference [73]. Given that the on-site observation is also included in these multiple sources, the total percentage of the methods that use the manual method of ground truth map acquisition is further increased to over 81%. One study, which corresponds to 9.1% of the total, does not mention the source of ground truth maps and another one uses street view images from Google Street View. No study reports whether ground truth map acquisition was done with the same timing or time period with the data collection.

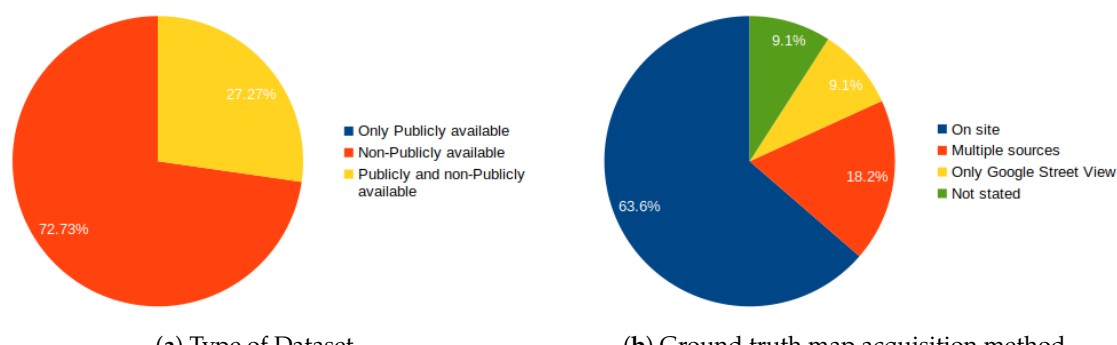

(**a**) Type of Dataset.　　　　　　　　(**b**) Ground truth map acquisition method.

**Figure 8.** Dataset: type, ground truth map acquisition source.

### 3.8. Cross-City Testing

Only 27.3% of the studies (three out of eleven studies) tested the cross city applicability of their proposed methods (Figure 9a). One did that [73] by applying the same heuristics in different cities. Carisi et al. [52] also applied heuristics for identifying SS and TL in different regions of California, from which two datasets had been collected. In both datasets, the accuracy of predictions is reported as over 90%. In both studies [52,73], testing for cross-city applicability is not explicitly declared. Only Saremi et al. [76] provide special analysis for this aspect of the problem. They trained a random forest classifier in city A and tested it in a city B (A and B belong to the same state). They report more than 92% accuracy of testing results. They also tested the case of training in city A of state A′ and testing in city B of state B′. They achieved an accuracy above 91%. Nevertheless, they point out that the generalization of their finding might not hold across different countries (train in a city from country A and testing in another city of country B), as further tests need to be done.

### 3.9. Number of Passes

Another important aspect of the problem is the number of samples that a classifier needs for optimizing the categorization task. This aspect, as can be seen in Figure 9b, is explored only by one study [52]. For a binary classification problem (TL, SS), they found out that 5 samples are needed for SS reliable detection and 7 for TL. We note that the classification method used in that study is heuristics-driven and not machine learning oriented.

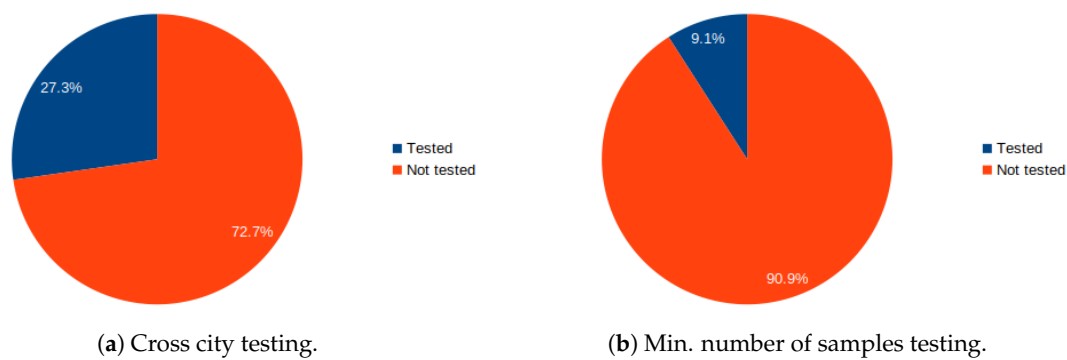

(**a**) Cross city testing.　　　　　　　　(**b**) Min. number of samples testing.

**Figure 9.** Tests applied for verification of cross city applicability and for the minimum number of samples needed by a classifier to perform sufficiently.

### 3.10. Classification Method

Figure 10 shows the different classification methods that the reviewed studies use. Four of them use decision trees (such as Random Forest) and an equal number interestingly use heuristics. By heuristics we mean a set of if-else rules that use threshold values for features that describe collectively the movement behaviour at traffic regulated locations. For example, Carisi et al. [52]

examine first each intersection for the case of being stop sign regulated, and, only if the probability of being a stop sign is below a threshold, they check for a traffic light. Due to missing or incomplete data, both heuristics might not be able to provide a clear categorization. In such cases they use some extra logical rules that governs the traffic regulated areas, by taking into account the possible rule combinations (e.g., an intersection is classified as regulated by a traffic light if at least half plus one of the incoming ways are marked as a potential traffic light). The movement behaviour at junctions is analysed in terms of two basic patterns—slow downs (speed less than 5 m/s within 50 m of an intersection) and standstills (speed less than 4 m/s within a time window of at least 10 sec and its extremes having at most 20 m spatial distance). An intersection way is regarded a *potential* stop sign regulated if at least 80% of the passes (traces) are standstills. If all or all but one ways belonging to a given intersection are marked as potential stops, the intersection is categorised as stop-sign regulated, and all the ways previously labelled as potential stops become actual stop signs. Similar heuristics use the other heuristic-based approaches shown in Table 10.

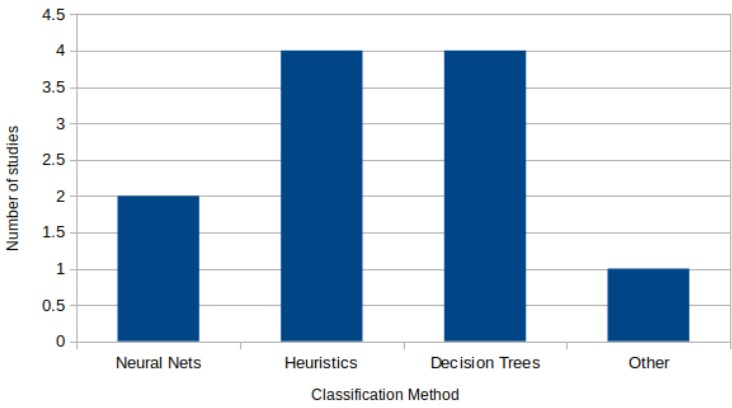

**Figure 10.** Classification methods used by the reviewed studies.

Two studies used Neural Networks and one does not report the classification method. Some studies state that they tested also additional classifiers but the results are not reported either because they did not differ much from the baseline classifier or they performed worse than the one that is documented in the article.

### 3.11. Classification Features

Among others, what makes a classifier powerful at identification tasks is the features it uses. Good features in general can make a classifier powerful, but certainly bad feature selection can make *any* classifier incapable of accomplishing even basic recognition tasks. The variety of features been used by the studies as shown in Table 10 is big. Nevertheless, all of them are related to the stop or slow down behaviour near intersections. How this behaviour is encoded in features differs from study to study. The non-heuristic methods use either statistical features that are estimated per intersection from the traversals that cross the latter (one study also use OSM map exported features) or other features related to the stop duration or the speed and acceleration (e.g., sequences of speeds [83]). The heuristic-driven approaches by conducting various tests, find thresholds for measures that can identify behaviours triggered by different regulators (e.g., the percentage of slowdowns and standstills at a SS location). For detecting TR, features again are related with the observed behaviour that is encoded either in form of percentage of turning instances [73] or towards/fromwards headings [79].

### 3.12. Classification Performance

The examined studies use different measures (Recall, Precision, Acuracy, etc.) for assessing their classification performance and therefore a quantitative analysis of the results of all studies was not

possible. Also, the composition of regulator categories examined by studies is the same only for three of the studies (TL, SS, RW), so even if performance measures were the same, comparisons could be done only for classification problems concerning the same composition of regulatory categories. However, their performance as shown in Table 10 seems promising, as almost all are over 80% accurate.

In the following section, all results are discussed under the framework of SLR research questions, which are listed in Table 1.

## 4. Discussion

*What:* Six categories of traffic regulators (TL, SS, RW, TR, YS, PS) were identified as possible to detect by crowdsensing means. A first observation on the results is that YS and PS are under-explored relative to other categories such as TL. Only one study [83] was found that dealt with these regulator categories, by formulating the problem as a binary classification task TL-YS/PS (low diversity). The reason for this fact may be related to the area of study that is interconnected with data availability. Judging only from the datasets that the studies use, one can see that, in Germany, it is very common for junctions (especially T-junctions) to be regulated by PS/YS, in contrast to in the USA where SS seems to be the alternative for the YS (all studies in USA involve SS and none YS/PS). Also, from personal experience from living in Germany, SS are rarely used as junction regulation means. This means that on a bigger dataset in Germany, SS could be also examined in the same categorization problem. This assumption might also hold for other countries too, since all these regulators are used on a bigger or a smaller scale in most countries. So the key to increasing the diversity of regulators within the same study might be a bigger dataset, so that instances from less popular regulators, such as SS are in Germany, have a less "outlier-like" arithmetic character in the dataset.

Another important finding is that cross-city applicability of classifiers is still an under-explored topic, given that only one study runs experiments for clarifying this aspect of the problem. Similarly, an under explored topic is the volume of data required for sufficient classification—5, 10, 20 or how many junction traversals are needed for a classifier to learn to distinguish different regulators?

However, the major finding of this SLR as can be deduced by the results is the importance of ground truth maps—84% of the studies rely on on-site observation for acquiring it and this no doubt imposes limitations on a method which is intended to automate a certain process. So, if by crowdsensing or using platforms that provide such data, lots of data can be readily available for dealing with the classification task, the acquisition of the ground-truth map is still needed for validation purposes. Nowadays, many GPS datasets are available from various open-source platforms or institutions or competitions but they cannot be used in the context of regulator detections unless the ground-truth map is also provided. Therefore, this finding emphasizes the need for open datasets that researchers can easily access, so that they do not have to come up with the time consuming manual work that a ground truth map involves. A possible way to construct such a ground-truth map could be the assignment of the task to the "crowd" and crowdsourcing of the traffic regulations.

*How:* Decision trees and heuristic rules were the most common methods being used in the studies. Heuristics in general require a manual or semi manual process for tuning the parameters (e.g., thresholds) that are then being used by rules derived most often by trial-error processes. In the context of the topic we examined, they provided results of equally high accuracy with the automatic approaches. One issue relevant to the threshold based approaches is whether the values of the found thresholds could provide equally successful results when applied to other cities or countries than those of the original studies. Yet no such testing is provided from the latter.

*How well:* More conclusions could be drawn if classification performance was assessed in all studies with the same metrics. Nevertheless, almost all the studies report high classification accuracy (over 80%), although the diversity of the examined categories is relatively low. For TR, only one study reports its performance (66–77%) so we cannot make generalisations for this regulator category.

*Other* A general observation already sketched in Section 1 is that crowd-sourcing traffic regulators for map-enrichment is a less explored topic compared to other automatically generated map elements

(road geometries, junction locations, etc.). This is reflected also from the number of articles excluded during the screening process as well as from the number of articles that makes up the under review list. The recent interest on the topic, if considered the long inactivity period since 1999 that the first relevant article was published, could be explained from the practical need to enhance maps with this information. This need comes from the recent progression of location-aware services, that in general try to optimize the transportation or to offer an optimal way to reach a location B from a location A, based on personalised or non-personalised criteria. No doubt, knowing intersection regulations can contribute to such optimizations.

## 5. Future Directions

So far, we have presented the different studies in a compact comparative way, highlighting what they actually do, how they do it and what they succeed at. What still might be not apparent are the future directions of this research area. We identify three major directions, some already underlined by the reviewed studies and others that resulted from this SLR. The first is the cross-city applicability as discussed in Section 3.8. It is still not clear whether a trained classifier could be applied across different cities or even countries and under which possible conditions, if any. Saremi and Abdelzahe [76] note that they are not convinced of generalising their promising cross-country results and suggest further experimentation on that topic. The second future research direction regards the regulator classification performance under highly diverse regulator categories. The studies so far examine the junction classification problem of at most three regulatory classes. More classes nevertheless reflect better the real world road network regulator system and certainly deserve a thorough examination. Last, a third future development should be directed to a hybrid GPS/computer-vision data approach where the GPS-based road regulation recognition result could be refined or opportunistically "assisted" from imagery data captured by camera. A weakness of the GPS-based methods is that they are highly dependent on a "dense" sampling of road ways, so that movement patterns can be detected from them. Very often it is the case where one junction, suppose a four-way junction, is well sampled in one of its ways and the three other ways are sparsely or not at all sampled. In such cases, a computer-vision approach could assist the classification by querying, for example, street level images from relatively new crowdsourced geotagged photo platforms, such as Mapillary [84].

## 6. Conclusions

Mapping with surveying equipment is a time-consuming and cost-intensive procedure, which means that practically it cannot be often repeated. This then raises the issue of how useful a map can be if it is not up-to-date. This SLR summarized the progression on detection and recognition of traffic regulators such as traffic lights (signals), stop-, yield-, priority-signs, right of way priority rules and turning restrictions at intersections, by leveraging non-imagery crowdsourced data. Non-imagery data are "lightweight" data in terms of data transfer and processing from the involved data collection participants and the data capture devices being used for the task. Another advantage of using GPS data over image-based data is that the former can be collected with minimum user involvement during the recording task, since no special equipment is needed (e.g., all current smartphones have GPS receivers) and no installation of the recording device is needed (e.g., camera) before starting the recording of the trip.

This SLR contribution is twofold—methods that can be used for regulation detection and recognition were located and summarized in a comparative way and research opportunities were emphasized after analysing each study (classification method) separately and in comparison with the others. The main limitation of the study is the fact that the performances of the different classification methods indeed cannot be compared as they do not reference the same dataset and performance metrics. Each study uses difference datasets with certain time, space and sample limitations (see Table 6) and the performance metrics used for assessing the classification accuracy are various. For this reason we

did not assess or interpret the methods' performance quantitatively, nor did we attempt to order them in any kind of order of accuracy or performance.

The two largest databases of peer-reviewed literature were used to locate eleven relevant studies. Thirteen aspects of the problem (dimensions) were selected for examination and information from each study was exported accordingly. That information was analysed in a semi-quantitative way, resulting in the following main findings—(1) regulator detection methods show high predictive ability across different categories, yet no study was found to examine all regulator categories within the same framework or at least to have a diversity over categories greater than 3; (2) only 27% of the studies examined the cross-city applicability and none relied solely on publicly available datasets. The minimum amount of data needed for classification (e.g., training) is also an under-explored aspect; (3) 81% of them acquired the ground truth map with on-site observation (in contrast to acquisition approaches relying on google street view images); (4) The accuracy of both heuristic-oriented approaches and common classification techniques provided in most regulator category settings an accuracy of over 80%. These major findings, combined with all the other issues being discussed in Sections 4 and 5, underline (5) the need for open datasets (with diverse traffic regulators) and ground truth maps that researchers can use as a benchmark for validating their methods and most importantly for comparing them with existing ones. Under the current progress in the field, no study can be reproduced for comparative purposes nor can it rely on open data sources due to lack of publicly available datasets and ground truth maps; (6) As a proposed future development, GPS-based traffic regulation inference could be opportunistically "assisted" from imagery data when, for example, the classification accuracy in certain locations is either low or when junctions are sparsely sampled by GPS tracks. In such cases, a vision-based approach such as traffic-sign recognition could be used to clarify the junction regulator context.

**Author Contributions:** Both authors contributed to conceptualization, methodology, investigation, validation, writing–review and editing and project administration. Stefania Zourlidou contributed to formal analysis, resources, writing–original draft preparation and visualization. Monika Sester contributed to supervision and funding acquisition.

**Funding:** This research was funded by the German Research Foundation (Deutsche Forschungsgemeinschaft (DFG)) with grant number 227198829/GRK1931. At the initial stage it was funded by IAV GmbH.

**Conflicts of Interest:** The authors declare no conflict of interest.The funders had no role in the design of the study; in the collection, analyses, or interpretation of data; in the writing of the manuscript, or in the decision to publish the results.

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
