# Peer review of "Traffic Regulator Detection and Identification from Crowdsourced Data—A Systematic Literature Review"

_ijgi, doi:10.3390/ijgi8110491_

Round 1

Reviewer 1 Report

The paper gives Systematic Literature Review (SLR) of the traffic regulator detection from crowdsourced data. It does not include any image analysis approach based on either aerial or satellite maps analysis or camera video analysis. It focuses solely on the papers which use GPS traces for traffic regulator detection.

The paper itself is well structured and informative. Judging by the years when the most of the reviewed papers were published, it also points out that this research area is in the spotlight. In that context the most important are Table 9 and Table 10.

My biggest problem with the paper is the section 2.1 Literature Search which is very broad and general. The explained steps can basically be applied to any SLR, not just the one devoted to the topic of traffic regulator detection. It makes the reader drift from the original topic of the paper. I am, therefore, suggesting that authors make this section shorter with down to the point explanations of the approach. It mostly applies to the first half of this section while the second half is relevant and related to the topic (eq. deciding on query terms).

The English language of the paper is good but there can be found sentences that are too long and sometimes hard to understand. In some cases just changing the order of the words would improve their meaning. Some examples are given in the reminder of this comment. I am suggesting to the authors to go though the paper again and try to split too long sentences.

At the end, I am suggesting couple of corrections that I was able to identify during the review:

line 61 - Road network consists of interconnected junctions which are geometrically complex locations and all of them are regulated with certain traffic regulators in shake of traffic participants’ safety - shoud be "sake" instead of "shake", can be split into two sentences, for example, in the following way:
Road network consists of interconnected junctions which are geometrically complex locations. Such junctions are regulated with certain traffic regulators in sake of traffic participants’ safety.

line 140 - Regarding the electonic databases, there are many different academic databases and search engines that someone can use for retrieving academic articles, such as ACM, IEEE Xplore, CiteSeer, DBLP, Google Scholar, JSTOR, SpringeLink, Science Direct, Scopus, Web of Science, Semantic Scholar, to mention a few relevant with the field of study we examine here. - should be "SpringerLink", too long and can be split into two sentences

line 146 - As explained later in the section 2.1.2, these databases provide advance search of documents based on personalised criteria and before querying them, special attention was given on carefully reading the search instructions and the various examples provided by the two service carriers. - should be "advanced search", also should be split for better readability

line 164 - should be "way too long time", or just "too long"

line 216 - Studies not directly related with the topic were excluded, like [65,66], which are more on traffic regulation violation detection or [67] that focuses on enriching maps for traffic sign compliance and others that although their method could be adopted for the objective of SLR, their focus although related to the SLR topic doesn’t coincide with it, such as [18,27,68]. - not understandable, should be split into more sentences

line 328 - should be "there is a long"
line 344 - Going back to the titles themselves, we can see that articles like [74] and [52] have quite of broad meaning titles, making indeed hard their association with the keyword search - change the order of the words to make the sentence more understandable
line 437 - should be "Four of them"

Author Response

Thank you very much for your comments and suggestions! Please see the attachment.

Reviewer 2 Report

The paper is generally well written and clearly argued. The topic reflects a desirable future direction for crowdsourcing which is an emerging topic and it should has received great attention from scholars and practitioners. Authors did a good job summarizing the results and looks interesting. The following are the comments that could be considered for authors while revising the paper:

The figure should be used correctly in the article. The place where the figure appears should be explained before the intention of the article (e.g. Figure 1). The title of table 2 is too long, authors should explain the contents of table 2 in the text. Future research should be placed in the chapters of conclusions and should include research limitations and contributions. Author do not need to put the abbreviation in the text for later explanation since it already explained in the text. There are too many subsections in the article, suggestions can be combined in the statement in the text (e.g. 2.3.1. 2.3.2…….). Sine traffic is a dynamic with peak and non-peak moments and the road may change slightly, this study used traffic regulator detection based on non-imagery, however, it is recommended that the author compare the accuracy of the results of this study with the results of other methods to highlight the contribution. In addition, much more attention needs to focus on describing the implications and contribution of the model and results to both research and practice.

Author Response

Thank you very much for the comments and suggestions! Please see attachment.
